# Association of Polyphenols Consumption with Risk for Gestational Diabetes Mellitus and Preeclampsia: A Systematic Review and Meta-Analysis

**DOI:** 10.3390/antiox11112294

**Published:** 2022-11-19

**Authors:** Gonzalo Jorquera, Romina Fornes, Gonzalo Cruz, Samanta Thomas-Valdés

**Affiliations:** 1Centro de Neurobiología y Fisiopatología Integrativa (CENFI), Instituto de Fisiología, Universidad de Valparaíso, Valparaíso 2360102, Chile; 2Centre for Translational Microbiome Research (CTMR), Department of Microbiology, Tumour and Cell Biology, Karolinska Institutet, Biomedicum Kvarter 8A, Tomtebodavägen 16, SE-171 65 Stockholm, Sweden; 3Escuela de Nutrición y Dietética, Facultad de Farmacia, Universidad de Valparaíso, Valparaíso 2360102, Chile; 4Centro de Micro-Bioinnovación, Universidad de Valparaíso, Valparaíso 2360102, Chile

**Keywords:** polyphenol-rich foods, gestational diabetes, preeclampsia, pregnancy comorbidity, observational studies

## Abstract

Gestational Diabetes Mellitus (GDM) and preeclampsia (PE) affects 6–25% of pregnancies and are characterized by an imbalance in natural prooxidant/antioxidant mechanisms. Due to their antioxidant and anti-inflammatory properties, polyphenols consumption during the pregnancy might exert positive effects by preventing GDM and PE development. However, this association remains inconclusive. This systematic review and metanalysis is aimed to analyze the association between polyphenol-rich food consumption during pregnancy and the risk of GDM and PE. A systematic search in MEDLINE, EMBASE, and Web of Science (Clarivate Analytics, London, United Kingdom) for articles dated between 1 January 1980 and July 2022 was undertaken to identify randomized controlled trials and observational studies evaluating polyphenol-rich food consumption and the risk of GDM and PE. The Newcastle-Ottawa Scale was used to evaluate the quality of these included studies. Twelve studies were included, of which eight articles evaluated GDM and four studied PE. A total of 3785 women presented with GDM (2.33%). No association between polyphenol consumption and GDM was found (ES = 0.85, 95% CI 0.71–1.01). When total polyphenol intake was considered, a lower likelihood to develop GDM was noted (ES = 0.78, 95% CI 0.69–0.89). Furthermore, polyphenol consumption was not associated with PE development (ES = 0.90, 95% CI 0.57–1.41). In conclusion, for both outcomes, pooled analyses showed no association with polyphenol-rich food consumption during pregnancy. Therefore, association of polyphenol intake with a decreased risk of GDM and PE remains inconclusive.

## 1. Introduction

Gestational Diabetes Mellitus (GDM) and preeclampsia (PE) are common medical complications during pregnancy, both showing a complex etiology, which is characterized by a genetic charge and a marked influence of lifestyle [1,2]. Depending on diagnostic criteria, among 6% to 25% of pregnant women can be affected by GDM [3,4]. This high prevalence, in turn, represents a higher risk of pregnancy-induced hypertension and PE [5].

The pathophysiology of GDM involves a multisystem insulin resistance which has several causes, including a genetic predisposition, an unhealthy lifestyle, and the action of some hormones produced during pregnancy [1]. The insulin resistance leads to an increase in blood glucose levels, allowing an excessive transport of glucose into the fetus through the placenta [6]. Moreover, insulin resistance leads to an increase in free fatty acids that can be transported in excess into the fetus [6]. Hyperglycemia during pregnancy is associated with an increased risk of suffering pregnancy complications but also impacts the development of the offspring [1,6]. The pathophysiology of preeclampsia involves an increase in peripheral vascular resistance, principally due a high resistance of placental blood vessels [2]. This leads to a decrease in the nutrients and oxygen supply to the fetus and increases the blood pressure of the mother, which is a risk factor for generating eclampsia and death [2]. At a molecular level, both GDM and PE conditions share an imbalance in the prooxidant/antioxidant mechanisms [7,8]. For example, in GDM pregnancies, an exaggerated production of reactive oxygen species (ROS) and placental oxidative stress, along with a disturbance in radical scavengers’ function, has been reported [7]. Moreover, circulating levels of antioxidants as selenium, zinc, and vitamin E, as well as serum total antioxidant capacity, were significantly lower in women with GDM than in normal pregnancies [9,10,11]. In the case of PE, pathological alterations are associated with oxidative and nitrosative free radical production by the placenta, which is considered to be one of the main molecular determinants of the disease during pregnancy [8].

Polyphenols are naturally occurring compounds in plants that have been recognized for their beneficial effects to human health [12]. Among the benefits, polyphenols have been demonstrated to improve glucose tolerance, improve redox status, reduce chronic low-grade inflammation, and reduce fatty liver [12,13,14,15]. Polyphenols can be classified as flavonoids and non-flavonoids, where the main sub-classes of flavonoids are flavanols, flavonols, anthocyanidins, flavones, flavanones, and chalcones, while the most studied non-flavonoids are stilbene, phenolic acids, saponins, and tannins [12]. Polyphenols have an antioxidant capacity that can be used against oxidative stress related diseases [13,16]. In this line, many studies have suggested that the ingestion of extracts enriched in polyphenols or polyphenol-rich foods may exert positive effects in maternal and fetus health during pregnancy, probably due to their antioxidant capacity and anti-inflammatory effects, although their real benefits are still controversial [17,18,19]. This systematic review and meta-analysis aims to analyze the effect of polyphenol-rich food consumption on the risk of GDM and PE. This study will shed light on the necessity of more studies on polyphenol’s effects on human pregnant women and will determine if it is necessary to include these substances in the nutritional guidelines, especially in women at risk of GDM or PE.

## 2. Methods

This systematic review and meta-analysis was based on an a priori established study protocol that followed the Cochrane Handbook guidelines for Systematic Reviews. The results are reported according to Preferred Reporting Items for Systematic Reviews and Meta-analyses (PRISMA) guidelines. PRISMA checklist is presented in Appendix A).

### 2.1. Search Strategy and Information Sources

The search strategy was previously prepared by the authors, which was curated and optimized by the Karolinska Institute University Library professionals (Figure 1). The electronic search included the revision of MEDLINE, EMBASE, and Web of Science (Clarivate Analytics) between 1 January 1980 and July 2021 and updated in September 2022, combining, between others, the following Medical Subject Headings (MeSH-terms) and keywords: ‘polyphenols sub-classes or polyphenol-rich foods’ and ‘gestational diabetes mellitus’ or ‘preeclampsia’. This search strategy allowed the identification of relevant studies reporting original data for the evaluation of polyphenol consumption and the risk of GDM and PE (Appendix A). Further, all the included articles were manually reviewed to identify any unidentified publications. No restrictions on language, method of publication, or geographical area were applied.

### 2.2. Eligibility Criteria and Study Selection

Firstly, all the studies identified by the systematic search were compiled in the reference management software ENDNOTE X20. The authors RF, ST, and GJ filtered and selected the articles, and GC solved discrepancies when needed. The articles included in the meta-analysis were based on the following eligibility criteria:(i)original studies from 1980 without language restrictions that assessed the polyphenol intake in pregnant women between the ages of 18–40 years and the risk of GDM or PE;(ii)studies of polyphenol-rich foods consumption by food frequency questionnaires or direct dietary intervention of polyphenols, curcumin, resveratrol, flavonoids, quercetin, tannins, catechins, phenolic acid, hydroxybenzoic acid, hydroxycinnamic acid, anthocyanins, or polyphenol-rich foods (tea, berries, chocolate, coffee);(iii)randomized controlled trials (RCTs), observational studies (prospective cohort studies);(iv)studies providing odds ratios (ORs), relative risks (RRs), or hazard ratios (HRs) along with 95% confidence intervals (CIs) or sufficient data to calculate the effect size (ES). When the article provided more than one estimator, the most adjusted one was selected.

The exclusion criteria were: (i) studies with no control of polyphenol-rich foods intake; (ii) case reports, abstracts, and reviews of animal or in vitro studies; (iii) narrative reviews, meta-analysis, systematic reviews, book chapters and conference abstracts.

### 2.3. Quality Assessment

The quality assessment of the included studies (cohort and case-control) was assessed by authors RF and ST using the Newcastle-Ottawa Scale (NOS) for observational studies and case-control studies. A NOS score ≥7 was considered of high quality (Appendix A). The disagreements were resolved via mutual consideration with author GJ. No studies were excluded due to low scores. A Meta-analysis of Observational Studies in Epidemiology (MOOSE) was incorporated in Appendix A).

### 2.4. Data Extraction

After checking and excluding studies with same population assessed, the data extraction included: (i) characteristics of the population (age and sample size, comparison group); (ii) data on trials (design, country of origin, duration of follow up, and risk of bias); (iii) exposure (polyphenols, polyphenol-rich foods evaluated in direct interventions or food frequency questionnaires); (iv) outcome measured in the studies (confounders assessed and the crude and adjusted estimates); (v) essential data including available raw data of incidence and proportions, cumulative incidence of the events of interest and HR with 95% CI.

### 2.5. Data Synthesis and Statistical Analysis

The pooled effect size (ES) was calculated by DerSimonian and Laird random effects meta-analysis and the Cochrane Q and I-squared statistics were used to assess the heterogeneity between the studies, I^2^ > 75% being indicative of high heterogeneity, 50–75% of moderate, and <50% of low heterogeneity. In case of Q test, *p* < 0.10 was considered indicative of heterogeneity. Supplementary analyses were conducted based on total polyphenol intake or their subclasses (i.e., anthocyanins, isoflavones, anthocyanins enriched fruits, and fruits). All statistical analyses were performed with Stata MP 17.0 (Stata Corp., College Station, TX, USA).

## 3. Results

The systematic search on polyphenols consumption and pregnancy complications included 4202 articles with publication dates until July 2021, uploaded in September 2022. After filtering based on titles and abstracts, 39 studies were included for a full assessment (Figure 1). Only eight studies were deemed eligible for risk assessment of GDM and four for PE (Table 1 and Table 2, respectively). Appendix A shows excluded studies. All selected studies were written in English and were published between 2007 and 2021. No new articles were found in the search conducted in September 2022.

### 3.1. Selected Studies

In studies assessing the association between polyphenol consumption and the risk of GDM, women were recruited from hospitals [24,26,27,30,31] or national cohort studies [25,28,29]. In articles assessing the risk of PE, three studies were hospital or care practices based [20,21,22] and one was a national cohort study [23]. Age and body mass index (BMI) of the participants in each article are shown by exposure, or by group of women affected (Table 1 and Table 2).

From the studies selected to evaluate the association of polyphenol-rich food consumption and GDM risk [24,25,26,27,28,29,30,31], the same population was evaluated in Dong et al. (2019), [28] and Dong et al. (2021) [29]. Therefore, two different metanalyses were performed including chocolate and soy isoflavone consumption, respectively (Figure 2A,B). In both cases, the analysis included a total of 162,443 individuals. All eight studies were cohort and corresponded to self-administrated questionnaires [24,26,28,29] or in-person/telephone interviews to apply questionnaires [25,27,30,31]. The majority of these studies were conducted in Asia [27,28,29,30,31], followed by Europe [25,26] and North America [24]. In addition, we analyzed separately studies that quantified total polyphenol intake or their subclasses (Appendix A). Polyphenol-rich food intake was estimated by using a food frequency questionnaire [24,28,29,30], 3-day 24 h dietary records or 4-day weighed food records [26,27,31], and other types of questionnaire [25]. Those articles that measured 3-day 24-h dietary records to estimate GDM risk evaluated the consumption of polyphenol-rich foods during each gestational trimester [27,31], whereas Tryggvadottir and co-authors [26] measured 4-day weighed food records during 19 to 24 gestational weeks. Some studies estimated the intake of polyphenol-rich foods 12 months prior with the participants recruited until delivery [28,29], others evaluated exposure before trying to get pregnant until delivery [24], the usual frequency consumption during the past 4 weeks [30], or during the pregnancy at gestational weeks 12 and 30, and after 6 and 18 months of delivery [25]. From these eight articles, three compared no consumption of polyphenol-rich foods with high consumption to estimate GDM risk [24,25,31], while others (five studies) evaluated low consumption of polyphenol-rich foods versus high consumption [26,27,28,29,30].

Four articles evaluated the polyphenols as polyphenols or their sub-classes [27,29,30,31], while other four articles assessed polyphenol-rich foods [24,25,26,28]. From these, the exposure was measured by evaluating the consumption of total polyphenol [27], soy isoflavone [29], total polyphenol, total flavonoids or total anthocyanins [30], total anthocyanins [31], coffee [24], coffee or tea [25], fruits and berries or coffee, tea and cocoa [26], and chocolate [28]. In addition, we analyzed separately those studies that quantified total polyphenol intake and PE association without gestational hypertension (Appendix A).

The evaluation of the association of polyphenol-rich food consumption during pregnancy and PE risk was based in only four studies that met the eligibility criteria [20,21,22,23]. Triche and co-authors [20] compared low consumption of polyphenol-rich foods vs. high consumption, whereas the other three articles compared no consumption with high consumption to estimate PE risk [21,22,23]. Kawanishi and co-authors [23] determined the polyphenol rich-food intake (tea or coffee) by using a questionnaire during the second or third trimester of pregnancy after recognizing conception, while other authors evaluated the risk to develop PE, estimating the intake of polyphenol-rich foods since women became pregnant until delivery [20,21,22]. In both cases, the analysis considered a total of 90,091 individuals. One study was retrospective [21] and three were cohort studies [20,22,23]. As the article from Kawanishi et al. (2021) [23] showed two estimators, tea and coffee consumption and the risk of PE, two meta-analyses were performed. An in-person interview with a questionnaire designed for the study was used for three studies [20,21,22], while just one used self-administrated questionnaires [23]. The food frequency questionnaire was used for just one study to determine polyphenol food consumption [23], while the others used self-designed questionnaires. These studies were conducted in North America [20,21,22] and Asia [23]. The information of the exposure corresponded to chocolate [20,22], or coffee or tea consumption [21,23]. The quality assessment of included studies is presented in Appendix A.

### 3.2. Gestational Diabetes Mellitus Risk Assessment

Figure 2A,B shows the analysis of the association between polyphenol exposure during pregnancy and the diagnosis of GDM. In the selected observational studies, the incidence of GDM was 2.39%. The pooled analysis showed no association between polyphenols exposure during pregnancy and GDM [2A: ES = 0.85 (95% CI 0.71–1.01), I^2^: 75.2%, *p* = 0.000, *n* = 7] and [2B: ES = 0.86 (0.73–1.01), I^2^: 69.8%, *p* < 0.003, *n* = 7] (Figure 2A and Figure 2B, respectively). The supplementary analysis showed that only quantified total polyphenol intake or their subclasses had significant results ES = 0.78 (0.69–0.89), I^2^: 0%, *p* < 0.37, *n* = 3] (Appendix A). Further, when anthocyanins-enriched fruits or polyphenols from fruits were analyzed, no association with GDM was found (Appendix A).

### 3.3. Preeclampsia Risk Assessment

In the selected observational studies, the incidence of PE was 2.78%. The analysis of polyphenols consumption with tea and chocolate as exposure was not associated with a risk of PE [ES  =  0.90 (0.57–1.41), I^2^: 69.9%, *p* < 0.019, N  = 4] (Figure 3A). This result was similar when chocolate and coffee were assessed [ES  =  0.74 (0.49–1.12), I^2^: 45.5%, *p* = 0.138, N = 4] (Figure 3B). A sub-analysis excluding Saftlas et al. was performed because they included gestational hypertension between study participants [22]. In this case, no association between polyphenol consumption and the risk of PE was found [ES  =  1.12 (0.97–1.29), I^2^: 0%, *p* = 0.532, N = 3] (Appendix A).

## 4. Discussion

### 4.1. Our Findings

No association was found between overall polyphenol consumption and the risk of GDM or PE in this meta-analysis of twelve studies enrolling 252,534 pregnant women. However, in subtypes analyses, a protective effect (RR = 0.78) was noted against GDM. Coffee or chocolate consumption was associated with a reduced risk of PE in two studies involving 88,073 pregnant women [22,23]. This might imply a possible protective effect of these nutrients that warrants further study. Gestational diabetes mellitus and pre-eclampsia are associated with increased risks of maternal and fetal adverse outcomes [2,32,33,34]. It is still unclear whether a specific diet might decrease the risk of developing GDM and PE. In a prior systematic review and meta-analysis, the risk of GDM was 0.57 (0.41–0.79) in those studies evaluating Mediterranean diet but not with polyphenol-rich foods intake [35]. However, the positive association might have been taken with caution. It because the mentioned study included cross-sectional reports, a majority of studies were based on self-questionnaires prone to recall bias, and the studies evaluated exposure to Mediterranean diet (and not proper polyphenols intake) [35]. Another systematic review involving 18 studies found that low-carbohydrate, ethnicity-based, and low glycemic index diet might be associated with better maternal glycemic control parameters [36]. No prior meta-analyses examining the risk of PE with polyphenols intake was found.

Both GDM and PE are characterized by an imbalance of oxidative stress in pregnant women [7,8]. It is necessary to improve the strategies to prevent these diseases. Adopting healthier lifestyle changes like healthy diets and increase physical activity has been associated with blood glucose control. However, whether polyphenols in particular have more benefit associated with lower risk is unknown/unclear. The potential role of polyphenol-rich foods against GDM and PE is described in Figure 4.

### 4.2. Why Is It Expected That Polyphenol-Rich Food Intake Protects against GDM?

During the last decades, a growing number of in vitro, preclinical and clinical studies showed a positive effect of polyphenols from fruits and vegetables on glucose homeostasis. The suggested mechanisms were as follows: inhibiting digestive enzymes; reducing activation of glucose transporter in small intestinal epithelial cells; decreasing the uptake of glucose into blood for limiting basolateral cell transport [37,38,39]; protecting pancreatic β-cells function; increasing insulin sensitivity [40,41]. All the mentioned mechanisms should lead to decreased basal and post-intake plasma glucose levels, leading to improved glucose tolerance. Evidence for improving glucose management has been observed for polyphenol rich fruits [42] and for resveratrol [43,44] in type 2 diabetic patients. Improving glucose management in pregnant women with GDM is fundamental to reduce their risk for developing pregnancy complications. For example, the recent metanalysis by Li X et al. [45] shows that exercise intervention improves glucose management and reduces adverse pregnancy outcomes such as premature birth and macrosomia. Similar improvements were observed under the treatment with insulin or metformin [46]. Selected fruits enriched in polyphenols, such as berries, have a great potential to improve glycemic management [42,47] and this improvement in glucose management should lead to improvements of pregnancy and neonatal outcomes.

On lipid metabolism, polyphenols demonstrated an increment of fatty acid oxidation associated to fat synthesis inhibition [41,48], improvement of mitochondrial functionality [49,50], reduction of lipid emulsification, inhibition of preadipocytes differentiation, and proliferation [51].

Improvements in lipid metabolism of pregnant mothers with GDM is expected to reduce the impact of GDM on offspring development, since maternal triglycerides and total cholesterol are independent risk factors for pregnancy complications, although other lipid metabolites should also be considered [52].

Another factor implicated in GDM is gut microbiota. Indeed, the abundance of some genera of bacteria is associated with glucose metabolism in GDM women [53] and gut microbiota could be mediating the increase in BMI of infants born to GDM women [54]. Interestingly, it is suggested that treatment with probiotics could improve insulin sensitivity, glycemic control, and pregnancy outcomes in GDM women [55,56,57]. In this line, treatment with polyphenols improves gut microbiota dysbiosis in both animal models and human studies, which is also related to the reduction of systemic inflammation that is common in GDM and PE [58,59]. Therefore, polyphenol-rich food consumption may improve gut dysbiosis in GDM women, thus improving glycemic control and pregnancy outcomes.

### 4.3. Why Is It Expected That Polyphenol-Rich Food Intake Protects against PE?

Polyphenols can exert benefits on blood pressure regulation by inducing vasodilation throughout two different pathways, which include increment of nitric oxide (NO) and vascular smooth muscular cells (VSMC) relaxation [60]. The antioxidant properties induced by polyphenols, especially flavonoids, are responsible for reducing reactive oxygen species being related to increased eNOS activity, which raise NO activity and content, whereas VSMC relaxation is mediated by inhibition of vascular Ca^2+^ channels and modulation of vascular BK_Ca_, endothelial IK_Ca,_ and SK_Ca_ channels [60]. Despite the wide evidence of beneficial effects of polyphenols on vascular function, it is unknown whether they might have a role in preventing preeclampsia.

### 4.4. Why Is It Important to Find New Strategies to Manage GDM and PE?

GDM increases the risk of metabolic disturbances for the newborn and perinatal outcomes, including congenital abnormalities, macrosomia or large for gestational age, stillbirth, prematurity, jaundice, hypoglycemia and/or hyperinsulinemia at birth, respiratory distress syndrome, shoulder dystocia, and higher risk of obesity, type 2 diabetes, and impaired glucose intolerance during adulthood [32,33,34].

The treatment of GDM is fundamental to prevent pregnancy complications and perinatal morbidity [61]. Moreover, we should go further and consider that improving glycemic control will protect the mother and their offspring from the long-term consequences triggered by GDM [62]. Currently, the treatments for GDM are lifestyle modifications and the use of insulin or metformin. Metformin and insulin treatments during GDM appear to be similar regarding the consequences in the mother [63] and their offspring [64], but they are the only two treatments for GDM.

PE is one of the most prevalent pregnancy-associated diseases which impacts both pregnancy outcomes and the newborn’s outcomes. Unfortunately, there is no other cure for preeclampsia than delivery. However, PE management is essential to decrease the impact of low nutrient and oxygen supply to the fetus and also prevent serious consequences of high blood pressure in the mother. In this sense, the use of resveratrol as a supplement for the treatment with nifedipine resulted in improving blood pressure control in patients with PE [65]. Moreover, plasma from patients consuming resveratrol or polyphenol-rich red grape juice decreases antioxidant markers and improves nitric oxide production in human umbilical vein endothelial cells (HUVEC) [66].

### 4.5. Strengths and Limitations

This is the first systematic review and metanalysis that evaluated the association of polyphenol-rich food subtype intake and the risk of PE. Moreover, it is the second metanalysis that evaluated the association of polyphenol intake and the risk of GDM. However, as discussed above, in the present metanalysis, we excluded articles concerned with the Mediterranean diet, which has other bioactive compound with healthy proprieties (i.e., n-3 unsaturated fatty acids, carotenoids and others). The advantage of our article is that we selected mainly prospective studies in which the diagnosis criteria were equivalent among articles. Here, we used studies in which the consumption of polyphenol rich foods was either estimated with food questionnaires or calculated, in which all included articles describe the follow-up and the exposure, in which the population was characterized, and in which control groups were used. In addition, we performed sub-analyses depending on polyphenol source and sub-classes of polyphenol.

Despite these, the metanalysis has several limitations including the heterogenicity of clinical studies regarding the source of polyphenol intake, the variability of questionnaires employed, and their accuracy in establishing a dosing of polyphenols consumed per day in the studied population. The lack of effects observed in both GDM and PE could be a result of the small number of studies and the in between studies’ heterogenicity. In addition, we could not ascertain the intake of polyphenols previous to pregnancy, which could add variability to the results of each study. Even if the clinical effects of polyphenols could be short term, it is expected that the benefits occur under a chronic consumption.

## 5. Conclusions

Overall consumption of a polyphenol-rich diet is not associated with decreased risk for GDM or PE. A beneficial effect might be present with coffee and chocolate consumption. The need for more prospective and/or controlled studies finding bioactive compounds that might prevent GM and PE is nevertheless warranted.

## Figures and Tables

**Figure 1 antioxidants-11-02294-f001:**
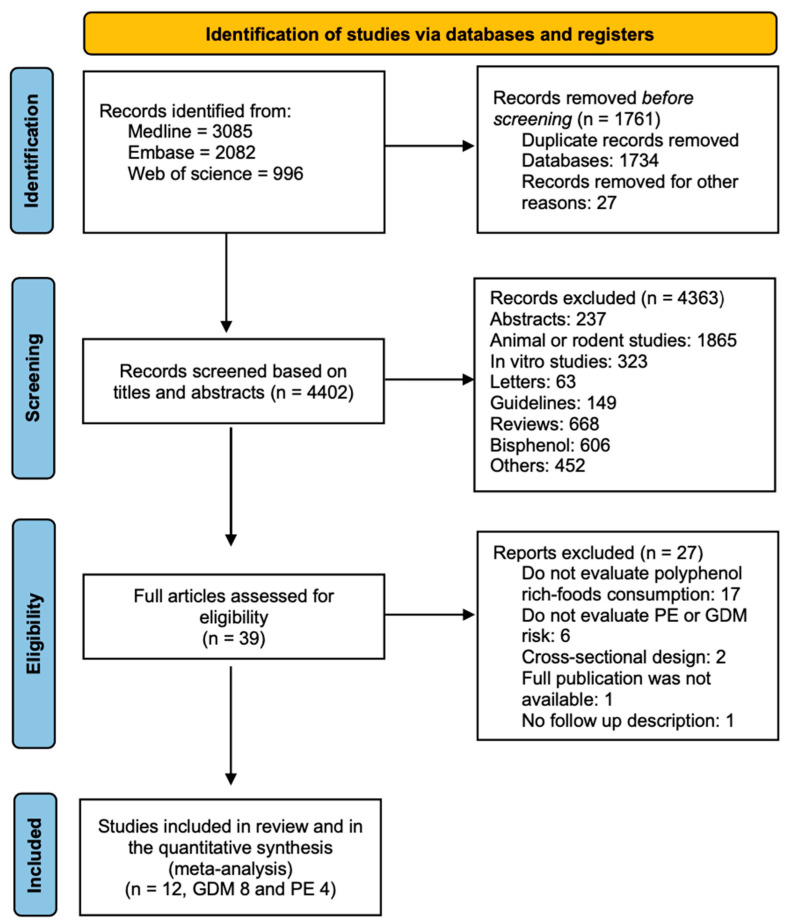
PRISM Flowchart for study selection. PRISMA, Preferred Reporting Items for Systematic Reviews and Meta-Analysis.

**Figure 2 antioxidants-11-02294-f002:**
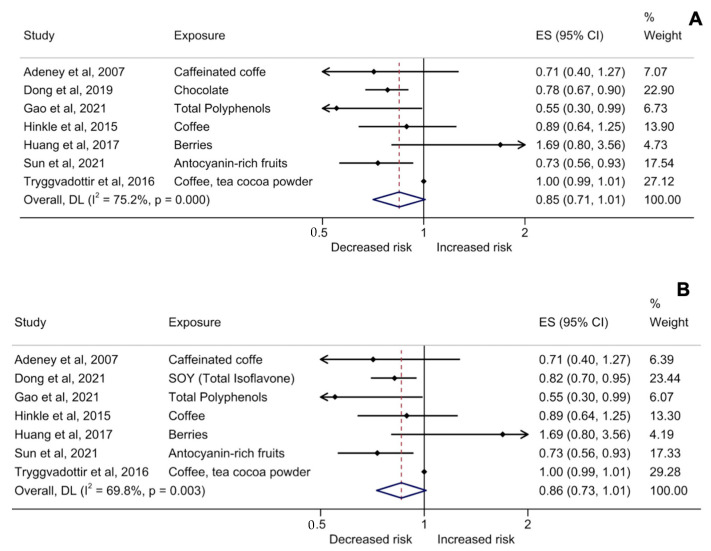
Forest plots for the pooled adjusted effect size of polyphenol-rich food consumption and risk for gestational diabetes mellitus in pregnant woman. Including Adeney et al., 2007 [24], Hinkle et al., 2015 [25], Tryggvadottir et al., 2016 [26], Huang et al., 2017 [27], Gao et al., 2021 [30], Sun et al., 2021 [31] plus in (**A**) Dong et al. (2019) [28] with chocolate intake; or in (**B**) including Dong et al. (2021) [29] with total isoflavone intake. Both studies analysed the same population. Abbreviations: CI = confidence interval; ES = effect size.

**Figure 3 antioxidants-11-02294-f003:**
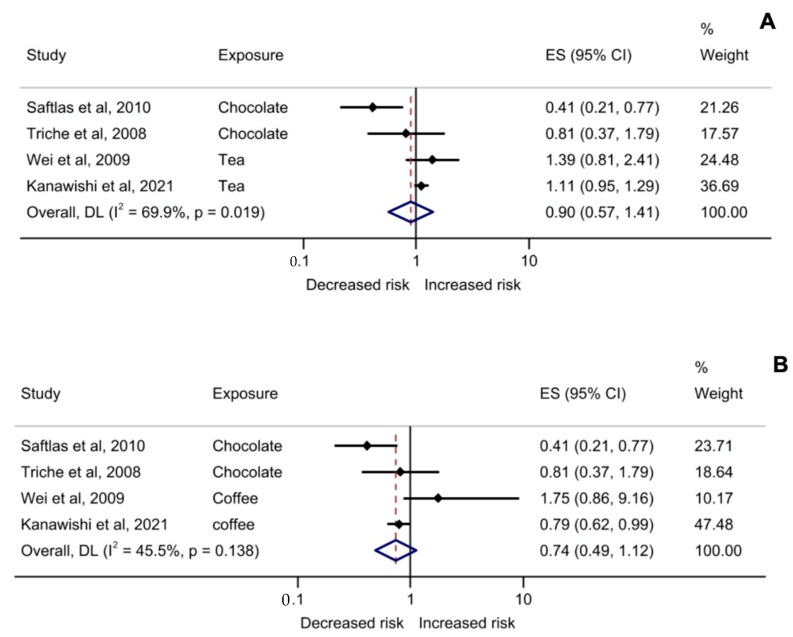
Forest plots for the pooled adjusted effect size of polyphenol-rich foods consumption and risk for preeclampsia in pregnant woman. Including Triche et al., 2008 [20], Saftlas et al., 2010 [22], and in (**A**) tea as polyphenol-rich food estimator for preeclampsia risk for Wei et al. (2009) [21] and Kanawishi et al. (2021) [23] was included; and in (**B**) coffee as polyphenol-rich food estimator for preeclampsia risk for Wei et al. (2009) [21] and Kanawishi et al. (2021) [23] was considered. Abbreviations: CI = confidence interval; ES = effect size.

**Figure 4 antioxidants-11-02294-f004:**
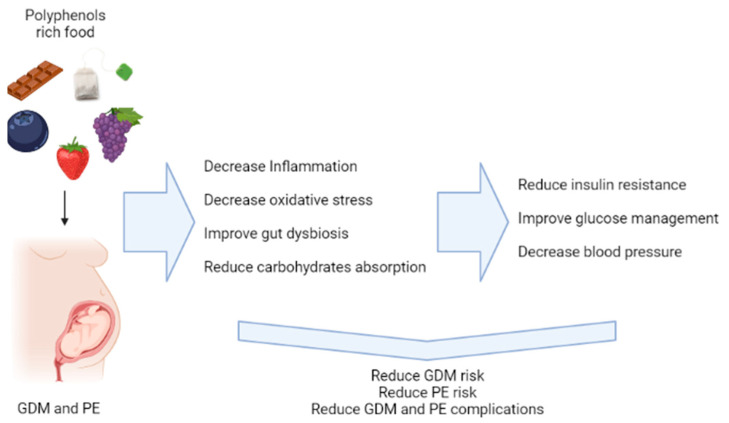
Scheme with potential role of polyphenol-rich foods against GDM and PE.

**Table 1 antioxidants-11-02294-t001:** Characteristics of studies on the polyphenol-rich food intake and risk of gestational diabetes mellitus and preeclampsia.

Author, Year, Country	Data Collection, Food Questionnaire	Study Design, Enrolment Period	Study Setting	Average Follow-Up (Weeks or Months)	PE */GDM Diagnosis (Cases)	Risk Estimate	Covariates Adjusted for **
**PE studies**							
[20] Triche et al., 2008, United States	In-person interview with a questionnaire designed for the study	Prospective, September 1996–January 2000	Multi-center, clinics and hospitals based	From 14 gestational weeks until delivery	Hypertension at least 6 h apart with proteinuria	Under 1 serving of chocolate vs. women 5 or more servings of chocolate per week during 1st trimester aOR = 0.81 (0.37, 1.79)	1st trimester smoking, clinic/private prenatal care provider, parity, race, education
[21] Wei et al., 2009, Canada	In-person interview with a questionnaire designed for the study	Retrospective, January 2003–March 2006	Multi-center, hospital based.	During pregnancy (not specified)	Hypertension at least 4 h apart with proteinuria	No tea drinker vs. tea persistent drinker during whole pregnancy aOR = 1.39 (0.81, 2.41) No coffee drinker vs. persistent drinker during whole pregnancy aOR = 175 (0.86, 9.16)	History of abortion, education, smoking
[22] Saftlas et al., 2010, United States	In-person interview with a questionnaire designed for the study	Prospective, April 1988–December 1991	Multi-center, care practices based	During pregnancy (not specified)	Hypertension at least 6 h apart with proteinuria	No regular chocolate consumption vs. 4 or more servings per week in 1st and 3rd trimester aOR = 0.41 (0.21, 0.77)	Parity, abortion history, maternal education, smoking during pregnancy, race, caffeine intake during pregnancy, fetal gender, GDM during pregnancy
[23] Kawanishi et al., 2021, Japan	Self-administered FFQ	Prospective, January 2011–March 2014	Nationwide birth cohort study	During pregnancy (not specified)	Hypertension with proteinuria	No coffee (Q1) vs. high coffee consumption (Q4 or ≥2 cups/day) aOR = 0.79 (0.62, 0.99) No tea (Q1) vs. high tea consumption (Q4 or ≥2 cups/day) aOR = 1.11 (0.95, 1.29)	Parity, pre-pregnancy smoking, alcohol consumption, folic acid supplementation, education, coffee and tea intake
**GDM studies**							
[24] Adeney et al., 2007, United States	Self-administered FFQ	Prospective, December 1996–September 2002	Multi center, hospital based	Gestational age 13 weeks at enrolment until delivery	1 h 50 g and 3 h 100 g oral glucose tolerance tests	None caffeinated coffee vs. high caffeinated coffee consumption (>7 cups per week) before pregnancy RR = 0.76 (0.40, 1.46)	Race, parity, smoking, alcohol, physical activity, employment during pregnancy, consumption of soft drinks, tea, decaffeinated coffee, caloric and fat intake, percentage of calories from fat, frequency of cream and sugar usage
[25] Hinkle et al., 2015, Denmark	In-person and telephone interviews with a designed for the study questionnaire	Prospective, March 1996–November 2002	National birth cohort study	From gestational ages of 12 and 30 weeks at enrolment until the child was 18 months of age	National Hospital Discharge Register and Post-delivery Interview	No tea intake vs. ≥8 teacups per day consumption RR = 0.77 (0.55, 1.08) No coffee intake vs. ≥8 coffee cups per day consumption RR = 0.89 (0.64, 1.25)	Parity, self-reported smoking status at the first interview, cola intake, socio-occupational status
[26] Tryggvadottir et al., 2016, Iceland	Self-performed 4-day weighed food record	Prospective, April 2012–October 2013	Single center, hospital based	From gestational age 19 to 38 weeks	75-g oral glucose tolerance test	Low vs. high tertile fruit and berries consumption aOR = 1.09 (0.99, 1.01) Low vs. high tertile coffee, tea and cocoa powder consumption aOR= 1.00 (0.99, 1.01)	Parity, energy intake, weekly weight gain, total metabolic equivalent of task
[27] Huang et al., 2017, China	In-person and telephone interviews for 3-day dietary record	Prospective. April 2013–August 2014	Single center, hospital based	From gestational age of 6 weeks at enrolment to mid/late pregnancy (≥28 weeks)	75-g oral glucose tolerance test	Low (Q1) vs. high (Q4) total polyphenols consumption (100 g/d of increment) aOR = 4.82 (2.39, 9.78) Low (Q1) vs. high (Q4) berries consumers (100 g/d of increment) aOR = 1.69 (0.80, 3.56)	Education, occupation, income, gestational weight gain, family history of diabetes, smoking and alcohol, consumption of grains, vegetables, meat and fish, glycemic index value of other fruit, consumption of other subtype of fruit
[28] Dong et al., 2019, Japan	Self-administered FFQ	Prospective, January 2011–March 2014	National birth, cohort study	Gestational age of 12 weeks at enrolment to 1 month after giving birth	75-g oral glucose tolerance test	Less than 1 servings of chocolate per month (Q1) vs. ≥7 times of chocolate per day (Q4) OR = 0.78 (0.67, 0.90)	Smoking, drinking education, occupation, depression, history of macrosomia babies, parity, physical activity, intake of total meat, red meat, coffee, green tea, milk, soya isoflavone, Mg, dietary fiber, dietary fat, saturated fat, snacks (potato chips or other crackers), total energy intake
[29] Dong et al., 2021, Japan	Self-administered FFQ	Prospective, January 2011–March 2014	National birth cohort study	From a median gestational age of 12 weeks at enrolment to mid/late pregnancy	75-g oral glucose tolerance test	Less than 1 servings of isoflavone product per month (Q1) vs. ≥7 times of isoflavone product per day (Q4) RR = 0.82 (0.70, 0.95)	Socio-demographic factors, disease history, medication, lifestyle factors, education level, history of depression, history of macrosomia babies, marital status, parity, smoking, drinking, physical activity, Western dietary pattern score
[30] Gao et al., 2021, China	In-person interview with FFQ	Prospective. January 2013–May 2016	Multi center, hospital based	Gestational age from 8 to 16 weeks at enrolment until delivery	75-g oral glucose tolerance test	Low (Q1) vs. high (Q4) total polyphenol consumers (100 g/d of increment) OR = 0.57 (0.30, 0.99) Low (Q1) vs. high (Q4) total anthocyanidin consumers (100 g/d of increment) OR = 0.62 (0.38, 1.00) Low (Q1) vs. high (Q4) total flavonoid consumers (100 g/d of increment) OR = 0.57 (0.32, 0.99)	1st and 2nd trimester weight gain, gravidity, parity, family history of diabetes, smoking and drinking status before pregnancy, physical activity, poor sleep quality, supplement use, dietary intake of vitamin C, vitamin E, fiber, cholesterol, selenium, zinc, and iron (all adjusted for energy intake), polyphenols from fruits and vegetables further adjusted nut polyphenols intake
[31] Sun et al., 2021, China	In-person and telephone interviews with a designed for the study questionnaire	Prospective, during 2017	Single center, hospital-based	40 weeks of pregnancy (information gathered each trimester)	75-g oral glucose tolerance test	Non consumers (Q1) vs. high consumers (Q4) of total fruits RR = 1.03 (0.83, 1.27) Non consumers (Q1) vs. high consumers (Q4) of total anthocyanin RR = 0.73 (0.56, 0.93)	Educational, family income, family history of diabetes, parity, smoking, alcohol, physical activity, energy, vegetables, whole grains, red meat, beverages, dietary fiber intake

* PE diagnosis: all studies defined hypertension as blood pressure ≥ 140 mm Hg systolic or ≥90 mm Hg diastolic after the 20th week of gestation and proteinuria as 24-h urine collection of ≥300 mg protein. ** Covariables adjusted for: all studies were adjusted for maternal age and pre-pregnancy body mass index (BMI), except Tryggvadottir et al. (2016), which used pre-pregnancy weight. Abbreviations: FFQ = Food frequency questionnaire; PE = Preeclampsia; BMI = body mass index; GDM = gestational diabetes mellitus; Q = quartile; RR = relative risk; OR = odds ratio; aOR = adjusted odds ratio.

**Table 2 antioxidants-11-02294-t002:** Participant characteristics for studies included in the meta-analysis.

Author, Year	Inclusion	Exclusion	Total Number of Patients	Number of Cases and Controls	Age, Years, Median or Mean	Pre-Gestational BMI	Multiparous, Primiparous	Ethnicity
**PE studies**								
[20] Triche et al., 2008	Pregnant who visited 56 obstetric practices and 15 clinics associated with 6 hospitals in Connecticut and Massachusetts.	Pregnant with more than 24 weeks’ gestational age at enrollment, with insulin-dependent diabetes mellitus, women that did not speak English or Spanish, or intended to terminate their pregnancy	1681	PE: 63 Controls:1618	#PE cases:29.0 ± 5.3 Control: 29.2 ± 5.0	#PE cases: 25.3 ± 3.4 Control: 23.7 ± 2.8	Both	Yes
[21] Wei et al., 2009	Nulliparous preeclamptic 48 hrs before delivery, at least 18 years of age, who spoke either French or English	Multiparous, had chronic hypertension or hypertension before 20 weeks of pregnancy, gestational hypertension without proteinuria, pregestational diabetes, heart disorders or HIV positive serology	337	PE: 92 Controls: 245	PE cases: 29.0 ± 5.2 Controls: 29.1 ± 5.3	PE cases: 23.9 ± 6.1 Control: 22.6 ± 4.2	Nulliparous	No
[22] Saftlas et al., 2010	Singleton pregnancy, women interviewed before 16 weeks of gestation, English speakers	Diabetes mellitus, non-English speaking, ≥16 weeks’ gestation or previous study participation	2540	PE: 58 GH: 158 Normal: 2324	#PE cases:30.7 ± 4.5 Control: 31.4 ± 4.5	Not clearly specified	Both	No
[23] Kawanishi et al., 2021	Singleton pregnancy	Multiple pregnancies, women with a medical history of hypertension, renal disease, history of HDP in previous pregnancies, and cases of DM and GDM	85,533	PE: 2222 Control:83,311	Lower quintile of caffeine intake: 31.1 ± 4.9 Higher quintile of caffeine intake 31.3 ± 5.1	Lower quintile of caffeine intake: 20.9 ± 3.0 Higher quintile of caffeine intake: 21.2 ± 3.3	Both	No
**GDM studies**								
[24] Adeney et al., 2007	NA	Patients were excluded when experienced a spontaneous or induced abortion, fetal demise prior to 28 weeks of gestation, those with prior insulin dependent or T2DM, interview data was missing or incomplete	1632	GDM: 75 Control: 1557	No coffee consumption: 31.7 ± 0.2 High coffee consumption: 33.5 ± 0.2	No coffee consumption: <20: 20% 20–24.9: 55% >25: 25% High coffee consumption <20: 19% 20–24.9: 54% >25: 27%	Both	Yes
[25] Hinkle et al., 2015	1st singleton pregnancy recorded in the register, women who completed the 1st two interviews	Pre-existing diabetes and deliveries if any relevant covariates were missing	71,239	GDM: 912 Control: 70,327	By exposure: Coffee: 0 cups/d: 29.2 ± 4.4 ≥ 8 cups/d: 31.7 ± 5.0 Tea: 0 cups/d: 29.5 ± 4.7 ≥8 cups/d: 31.3 ± 4.8	#By exposure: Coffee: 0 cups/d: 23.8 ± 4.1 8 cups/d: 23.7 ± 4.0 Tea: 0 cups/d: 24.0 ± 4.2 ≥8 cups/d: 23.6 ± 3.9	Nulliparous	No
[26] Tryggvadottir et al., 2016	Women living in Reykjavik, 18 and 40 years old, non-smokers with or without family history of diabetes or GDM, BMI above of 18.5	Parity >3	168	GDM:17 Control: 151	Normal weight: 29.0 ± 4.8 Overweight: 30.0 ± 4.3 Obese: 30.0 ± 4.6	Normal weight: 21.6 ± 1.6 Overweight: 27.2 ± 1.2 Obese: 33.2 ± 2.7	Multiparous	No
[27] Huang et al., 2017	Primiparous women, 20 to 35 years old at 6–12 weeks of gestational age, measurement of blood glucose during 24 to 28 gestation weeks	Multiparous, no information of blood glucose or lost to follow-up, abortion, multiple pregnancy, type 1 or 2 diabetes, hypertension, renal insufficiency, kidney stones, thyroid-gland dysfunction, chronic obstructive pulmonary disease or asthma, HIV infection or active tuberculosis virus, mental disorders or anemia	772	GDM: 169 Control: 603	Overall (general fruit consumption) Q1: 25.73 ± 3.11 Q4: 26.73 ± 3.3	Overall (General fruits consumption) Q1: < 18.5: 40.2% 18.5–24: 55.2% 24 or more: 4.6% Q4: = <18.5: 29.5% 18.5–24: 64.2% 24 or more: 6.2%	Primiparous	No
[28] Dong et al., 2019	Singleton pregnancy	Extreme BMI, history of stroke, heart disease, cancer, type 1 and/or 2 diabetes, GDM at study enrolment	84,948	GDM: 1904 Control: 83,044	Lowest chocolate consumption (Q1): 30.9 ± 5.1 Highest chocolate consumption (Q4): 30.5 ± 4.9	Lowest chocolate consumption (Q1): 21.3 ± 3.3 Highest chocolate consumption (Q4): 21.0 ± 3.0	Multiparous and nulliparous	No
[29] Dong et al., 2021	Singleton pregnancy, free of GDM, stroke, heart disease, Kawasaki disease, cancer, type 1 and/or 2 diabetes	Extreme BMI before pregnancy, extreme total energy intake (higher or lower)	84,948	GDM: 1904 Control: 83,044	Q1: 29.8 (5.3) Q5: 31.3 (4.9)	Q1: 21.4 (3.3) Q5: 21.0 (3.1)	Both	No
[30] Gao et al., 2021	Singleton pregnancy, age from 18 to 45 years, gestational age from 8 to 16 weeks	Blank items > 10 on FFQ, missing values for any vegetables or fruits, extreme energy intake, unavailable OGTT data or OGTT performed before FFQ, multiple pregnancies, pre-gestational diabetes	2231	GDM: 185 Control: 2046	Q1: 28.2 ± 3.4 Q4: 28.0 ± 3.5	Q1: 21.0 ± 2.6 Q4: 20.6 ± 2.5	Both	Yes
[31] Sun et al., 2021	Singleton pregnancy, gestational age from 6 to 14 weeks, no chronic metabolic disease	Missing data from the dietary recall or the OGTT, GDM history, extreme total energy intake (higher or lower)	1453	GDM: 523 Control: 930	Q1: 28.8 Q4: 28.4	Q1: 21.1 Q4: 20.4	Both	No

# When the upper or lower limit of the upper or lower interval was not specified (i.e., >40), the mean ± SD of age and BMI was calculated by using the lower or upper limit written in the article (i.e., if the highest interval in BMI was >40, the lower and the upper limit of that specific interval was 40). Abbreviations: GDM = gestational diabetes mellitus; PE = preeclampsia; OGTT = oral glucose tolerance test; Q = quartile; BMI = body mass index; d = day; hrs = hours; FFQ = food frequency questionnaire; HIV = human immunodeficiency virus.

## Data Availability

Not applicable.

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
