# Peer review of "Association of Polyphenols Consumption with Risk for Gestational Diabetes Mellitus and Preeclampsia: A Systematic Review and Meta-Analysis"

_antioxidants, 2022, doi:10.3390/antiox11112294_

Round 1
Reviewer 1 Report
The purpose of this manuscript is to analyse the association between the consumption of foods rich in polyphenols during pregnancy and the risk of developing Gestational Diabetes Mellitus or Preeclampsia. The scientific collect is very interesting, however, some problems, as indicated below, should be addressed before the document can be considered for publication in this journal. This version of the manuscript is not enough complete.
Minor revision:
English language and style are fine, minor spell check is required to ensure that an international audience can clearly understand your text. In general, I suggest reviewing the style of the manuscript according to the guidelines of the journal.
I suggest to the authors to introduce a paragraph in which the pathophysiological features of the two diseases are better explained.
Line 61: I would suggest that the authors give some examples of the positive effects of polyphenols on human health.
Line 68: I might suggest adding the following more recent references: 10.3389/fphar.2022.1015835; 10.3390/cells11152391; 10.3390/nu14204317.
Line 322: the acronym “TGs” was not written in full.
Author Response
Dear reviewer,
Thank you very much for all your comments. Please see the attachment with reply to your suggestions.
Best regards,
Samanta

Reviewer 2 Report
Gestational Diabetes Mellitus (GDM) and preeclampsia (PE) are characterized by an imbalance in natural prooxidant/antioxidant mechanisms. Polyphenols consumption during the pregnancy might exert positive effects by preventing GDM and PE development.
In this systematic review and metanalysis the authors aimed to analyze the association between polyphenol rich foods consumption during pregnancy and the risk of GDM and PE. The authors concluded that , for both outcomes, pooled analyses showed no association with polyphenol rich-foods consumption during pregnancy. Therefore, association of polyphenol intake with a decreased risk of GDM and PE remains inconclusive.
This is an interesting analysis approaching a relevant issue.
I only have some minor concerns.
Please add a schematic representation of the potential role of polyphenols in GDM and PE.
Please enlarge the section entitled “Why is expected that polyphenol-rich food intake protects against GDM? “ This is a very relevant issue for the Journal and deserve a larger analysis.
Author Response
Dear reviewer,
Thank you for all your comments. Please see the attachment with reply to your suggestions.
Best regards,
Samanta
